# Missing Not at Random in Matrix Completion: The Effectiveness of Estimating Missingness Probabilities Under a Low Nuclear Norm Assumption

**Wei Ma**[*]    **George H. Chen**[*]
Carnegie Mellon University
Pittsburgh, PA 15213
{weima,georgechen}@cmu.edu

## Abstract

Matrix completion is often applied to data with entries missing not at random (MNAR). For example, consider a recommendation system where users tend to only reveal ratings for items they like. In this case, a matrix completion method that relies on entries being revealed at uniformly sampled row and column indices can yield overly optimistic predictions of unseen user ratings. Recently, various papers have shown that we can reduce this bias in MNAR matrix completion if we know the probabilities of different matrix entries being missing. These probabilities are typically modeled using logistic regression or naive Bayes, which make strong assumptions and lack guarantees on the accuracy of the estimated probabilities. In this paper, we suggest a simple approach to estimating these probabilities that avoids these shortcomings. Our approach follows from the observation that missingness patterns in real data often exhibit low nuclear norm structure. We can then estimate the missingness probabilities by feeding the (always fully-observed) binary matrix specifying which entries are revealed or missing to an existing nuclear-norm-constrained matrix completion algorithm by Davenport et al. [2014]. Thus, we tackle MNAR matrix completion by solving a different matrix completion problem first that recovers missingness probabilities. We establish finite-sample error bounds for how accurate these probability estimates are and how well these estimates debias standard matrix completion losses for the original matrix to be completed. Our experiments show that the proposed debiasing strategy can improve a variety of existing matrix completion algorithms, and achieves downstream matrix completion accuracy at least as good as logistic regression and naive Bayes debiasing baselines that require additional auxiliary information.

## 1   Introduction

Many modern applications involve partially observed matrices where entries are missing not at random (MNAR). For example, in restaurant recommendation, consider a ratings matrix $X \in (\mathbb{R} \cup \{\star\})^{m \times n}$ where rows index users and columns index restaurants, and the entries of the matrix correspond to user-supplied restaurant ratings or "$\star$" to indicate "missing". A user who is never in London is unlikely to go to and subsequently rate London restaurants, and a user who is vegan is unlikely to go to and rate restaurants that focus exclusively on meat. In particular, the entries in the ratings matrix are not revealed uniformly at random. As another example, in a health care context, the partially observed matrix $X$ could instead have rows index patients and columns index medically relevant

---

[*]Equal contribution

Code available at https://github.com/georgehc/mnar_mc

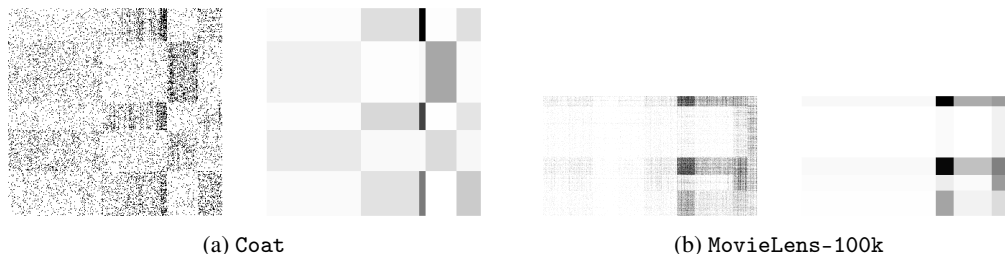

(a) `Coat`    (b) `MovieLens-100k`

Figure 1: Missingness mask matrices (with rows indexing users and columns indexing items) exhibit low-rank block structure for the (a) `Coat` and (b) `MovieLens-100k` datasets. Black indicates an entry being revealed. For each dataset, we show the missingness mask matrix on the left and the corresponding block structure identified using spectral biclustering [Kluger et al., 2003] on the right; rows and columns have been rearranged based on the biclustering result.

measurements such as latest readings from lab tests. Which measurements are taken is not uniform at random and involve, for instance, what diseases the patients have. Matrix completion can be used in both examples: predicting missing entries in the recommendation context, or imputing missing features before possibly using the imputed feature vectors in a downstream prediction task.

The vast majority of existing theory on matrix completion assume that entries are revealed with the same probability independently (e.g., Candès and Recht [2009], Cai et al. [2010], Keshavan et al. [2010a,b], Recht [2011], Chatterjee [2015], Song et al. [2016]). Recent approaches to handling entries being revealed with nonuniform probabilities have shown that estimating what these entry revelation probabilities can substantially improve matrix completion accuracy in recommendation data [Liang et al., 2016, Schnabel et al., 2016, Wang et al., 2018a,b, 2019]. Specifically, these methods all involve estimating the matrix $P \in [0,1]^{m \times n}$, where $P_{u,i}$ is the probability of entry $(u,i)$ being revealed for the partially observed matrix $X$. We refer to this matrix $P$ as the *propensity score matrix*. By knowing (or having a good estimate of) $P$, we can debias a variety of existing matrix completion methods that do not account for MNAR entries [Schnabel et al., 2016].

In this paper, we focus on the problem of estimating propensity score matrix $P$ and examine how error in estimating $P$ impacts downstream matrix completion accuracy. Existing work [Liang et al., 2016, Schnabel et al., 2016, Wang et al., 2018b, 2019] typically models entries of $P$ as outputs of a simple predictor such as logistic regression or naive Bayes. In the generative modeling work by Liang et al. [2016] and Wang et al. [2018b], $P$ is estimated as part of a larger Bayesian model, whereas in the work by Schnabel et al. [2016] and Wang et al. [2019] that debias matrix completion via inverse probability weighting (e.g., Imbens and Rubin [2015]), $P$ is estimated as a pre-processing step.

Rather than specifying parametric models for $P$, we instead hypothesize that in real data, $P$ often has a particular low nuclear norm structure (precise details are given in Assumptions A1 and A2 in Section 3; special cases include $P$ being low rank or having clustering structure in rows/columns). Thus, with enough rows and columns in the partially observed matrix $X$, we should be able to recover $P$ from the missingness mask matrix $M \in \{0,1\}^{m \times n}$, where $M_{u,i} = \mathbb{1}\{X_{u,i} \neq \star\}$. For example, for two real datasets `Coat` [Schnabel et al., 2016] and `MovieLens-100k` [Harper and Konstan, 2016], their missingness mask matrices $M$ (note that these are always fully-observed) have block structure, as shown in Figure 1, suggesting that they are well-modeled as being generated from a low rank $P$; with values of $P$ bounded away from 0, such a low rank $P$ is a special case of the general low nuclear norm structure we consider. In fact, the low rank missingness patterns of `Coat` and `MovieLens-100k` can be explained by topic modeling, as we illustrate in Appendix A.

We can recover propensity score matrix $P$ from missingness matrix $M$ using the existing 1-bit matrix completion algorithm by Davenport et al. [2014]. This algorithm, which we refer to as 1BITMC, solves a convex program that amounts to nuclear-norm-constrained maximum likelihood estimation. We remark that Davenport et al. developed their algorithm for matrix completion where entries are missing independently with the same probability and the revealed ratings are binary. We intentionally apply their algorithm instead to the matrix $M$ of binary values for which there are no missing entries. Thus, rather than completing a matrix, we use 1BITMC to denoise $M$ to produce propensity score matrix estimate $\widehat{P}$. Then we use $\widehat{P}$ to help debias the actual matrix completion problem that we care about: completing the original partially observed matrix $X$.

Our contributions are as follows:

- We establish finite-sample bounds on the mean-squared error (MSE) for estimating propensity score matrix $P$ using 1BITMC and also on its debiasing effect for standard MSE or mean absolute error (MAE) matrix completion losses (the debiasing is via weighting entries inversely by their estimated propensity scores).

- We empirically examine the effectiveness of using 1BITMC to estimate propensity score matrix $P$ compared to logistic regression or naive Bayes baselines. In particular, we use the estimated propensity scores from these three methods to debias a variety of matrix completion algorithms, where we find that 1BITMC typically yields downstream matrix completion accuracy as good as or better than the other two methods. The 1BITMC-debiased variants of matrix completion algorithms often do better than their original unmodified counterparts and can outperform some existing matrix completion algorithms that handle nonuniformly sampled data.

## 2 Model and Algorithms

**Model.** Consider a signal matrix $S \in \mathbb{R}^{m \times n}$, a noise matrix $W \in \mathbb{R}^{m \times n}$, and a propensity score matrix $P \in [0,1]^{m \times n}$. All three of these matrices are unknown. We observe the matrix $X \in (\mathbb{R} \cup \{\star\})^{m \times n}$, where $X_{u,i} = S_{u,i} + W_{u,i}$ with probability $P_{u,i}$, independent of everything else; otherwise $X_{u,i} = \star$, indicating that the entry is missing. We denote $\Omega$ to be the set of entries that are revealed (i.e., $\Omega = \{(u,i) : u \in [m], i \in [n] \text{ s.t. } X_{u,i} \neq \star\}$), and we denote $X^* := S + W$ to be the noise-corrupted data if we had observed all the entries. Matrix completion aims to estimate $S$ given $X$, exploiting some structural assumption on $S$ (e.g., low nuclear norm, low rank, a latent variable model).

**Debiasing matrix completion with inverse propensity scoring.** Suppose we want to estimate $S$ with low mean squared error (MSE). If no entries are missing so that we directly observe $X^*$, then the MSE of an estimate $\widehat{S}$ of $S$ is

$$L_{\text{full MSE}}(\widehat{S}) := \frac{1}{mn} \sum_{u=1}^{m} \sum_{i=1}^{n} (\widehat{S}_{u,i} - X^*_{u,i})^2.$$

However, we actually observe $X$ which in general has missing entries. The standard approach is to instead use the observed MSE:

$$L_{\text{MSE}}(\widehat{S}) := \frac{1}{|\Omega|} \sum_{(u,i) \in \Omega} (\widehat{S}_{u,i} - X_{u,i})^2.$$

If the probability of every entry in $X$ being revealed is the same (i.e., the matrix $P$ consists of only one unique nonzero value), then the loss $L_{\text{MSE}}(\widehat{S})$ is an unbiased estimate of the loss $L_{\text{full MSE}}(\widehat{S})$. However, this is no longer guaranteed to hold when entries are missing with different probabilities. To handle this more general setting, we can debias the loss $L_{\text{MSE}}$ by weighting each observation inversely by its propensity score, a technique referred to as inverse propensity scoring (IPS) or inverse probability weighting in causal inference [Thompson, 2012, Imbens and Rubin, 2015, Little and Rubin, 2019, Schnabel et al., 2016]:

$$L_{\text{IPS-MSE}}(\widehat{S}|P) := \frac{1}{mn} \sum_{(u,i) \in \Omega} \frac{(\widehat{S}_{u,i} - X_{u,i})^2}{P_{u,i}}. \tag{1}$$

Assuming $P$ is known, the IPS loss $L_{\text{IPS-MSE}}(\widehat{S}|P)$ is an unbiased estimate for $L_{\text{full MSE}}(\widehat{S})$.[2]

Any matrix completion method that uses the naive MSE loss $L_{\text{MSE}}$ can then be modified to instead use the unbiased loss $L_{\text{IPS-MSE}}$. For example, the standard approach of minimizing $L_{\text{MSE}}$ with nuclear norm regularization can be modified where we instead solve the following convex program:

$$\widehat{S} = \arg\min_{\Gamma \in \mathbb{R}^{m \times n}} L_{\text{IPS-MSE}}(\Gamma|P) + \lambda\|\Gamma\|_*, \tag{2}$$

where $\lambda > 0$ is a user-specified parameter, and $\|\cdot\|_*$ denotes the nuclear norm. Importantly, using the loss $L_{\text{IPS-MSE}}$ requires either knowing or having an estimate for the propensity score matrix $P$.

Instead of squared error, we could look at other kinds of error such as absolute error, in which case we would consider MAE instead of MSE. Also, instead of nuclear norm, other regularizers could be used in optimization problem (2). Lastly, we remark that the inverse-propensity-scoring loss $L_{\text{IPS-MSE}}$ is not the only way to use propensity scores to weight. Another example is the Self-Normalized Inverse Propensity Scoring (SNIPS) estimator [Trotter and Tukey, 1956, Swaminathan and Joachims, 2015], which replaces the denominator term $mn$ in equation (1) by $\sum_{(u,i)\in\Omega}\frac{1}{P_{u,i}}$. This estimator tends to have lower variance than the IPS estimator but incurs a small bias [Hesterberg, 1995].

For ease of analysis, our theory focuses on debiasing with IPS. Algorithmically, for a given $P$, whether one uses IPS or SNIPS for estimating $S$ in optimization problem (2) does not matter since they differ by a multiplicative constant; tuning regularization parameter $\lambda$ would account for such constants. In experiments, for reporting test set errors, we use SNIPS since IPS can be quite sensitive to how many revealed entries are taken into consideration.

**Estimating the propensity score matrix.** We can estimate $P$ based on the missingness mask matrix $M \in \{0, 1\}^{m \times n}$, where $M_{u,i} = \mathbb{1}\{X_{u,i} \neq \star\}$. Specifically, we use the nuclear-norm-constrained maximum likelihood estimator proposed by Davenport et al. [2014] for 1-bit matrix completion, which we refer to as 1BITMC. The basic idea of 1BITMC is to model $P$ as the result of applying a user-specified link function $\sigma : \mathbb{R} \to [0, 1]$ to each entry of a parameter matrix $A \in \mathbb{R}^{m \times n}$ so that $P_{u,i} = \sigma(A_{u,i})$; $\sigma$ can for instance be taken to be the standard logistic function $\sigma(x) = 1/(1 + e^{-x})$. Then we estimate $A$ assuming that it satisfies nuclear norm and entry-wise max norm constraints, namely that
$$A \in \mathcal{F}_{\tau,\gamma} := \big\{\Gamma \in \mathbb{R}^{m \times n} : \|\Gamma\|_* \leq \tau\sqrt{mn}, \ \|\Gamma\|_{\max} \leq \gamma\big\},$$
where $\tau > 0$ and $\gamma > 0$ are user-specified parameters. Then 1BITMC is given as follows:

1. Solve the constrained Bernoulli maximum likelihood problem:
$$\widehat{A} = \arg\max_{\Gamma \in \mathcal{F}_{\tau,\gamma}} \sum_{u=1}^{m} \sum_{i=1}^{n} [M_{u,i} \log \sigma(\Gamma_{u,i}) + (1 - M_{u,i}) \log(1 - \sigma(\Gamma_{u,i}))]. \tag{3}$$
   For specific choices of $\sigma$ such as the standard logistic function, this optimization problem is convex and can, for instance, be solved via projected gradient descent.

2. Construct the matrix $\widehat{P} \in [0, 1]^{m \times n}$, where $\widehat{P}_{u,i} := \sigma(\widehat{A}_{u,i})$.

## 3 Theoretical Guarantee

For $\widehat{P}$ computed via 1BITMC, our theory bounds how close $\widehat{P}$ is to $P$ and also how close the IPS loss $L_{\text{IPS-MSE}}(\widehat{S}|\widehat{P})$ is to the fully-observed MSE $L_{\text{full MSE}}(\widehat{S})$. We first state our assumptions on the propensity score matrix $P$ and the partially observed matrix $X$. As introduced previously, 1BITMC models $P$ via parameter matrix $A \in \mathbb{R}^{m \times n}$ and link function $\sigma : \mathbb{R} \to [0, 1]$ such that $P_{u,i} = \sigma(A_{u,i})$. For ease of exposition, throughout this section, we take $\sigma$ to be the standard logistic function: $\sigma(x) = 1/(1 + e^{-x})$. Following Davenport et al. [2014], we assume that:

**A1.** $A$ has bounded nuclear norm: there exists a constant $\theta \in (0, \infty)$ such that $\|A\|_* \leq \theta\sqrt{mn}$.

**A2.** Entries of $A$ are bounded in absolute value: there exists a constant $\alpha \in (0, \infty)$ such that $\|A\|_{\max} := \max_{u\in[m],i\in[n]} |A_{u,i}| \leq \alpha$. In other words, $P_{u,i} \in [\sigma(-\alpha), \sigma(\alpha)]$ for all $u \in [m]$ and $i \in [n]$, where $\sigma$ is the standard logistic function.

As stated, Assumption A2 requires probabilities in $P$ to be bounded away from both 0 and 1. With small changes to 1BITMC and our theoretical analysis, it is possible to allow for entries in $P$ to be 1, i.e., propensity scores should be bounded from 0 but not necessarily from 1. We defer discussing this setting to Appendix C as the changes are somewhat technical; the resulting theoretical guarantee is qualitatively similar to our guarantee for 1BITMC below.

Assumptions A1 and A2 together are more general than assuming that $A$ has low rank and has entries bounded in absolute value. In particular, when Assumption A2 holds and $A$ has rank $r \in (0, \min\{m, n\}]$, then Assumption A1 holds with $\theta = \alpha\sqrt{r}$ (since $\|A\|_* \leq \sqrt{r}\|A\|_F \leq \sqrt{rmn}\|A\|_{\max} \leq \alpha\sqrt{rmn}$, where $\|\cdot\|_F$ denotes the Frobenius norm). Note that a special case of $A$ being low rank is $A$ having clustering structure in rows, columns, or both. Thus, our theory also covers the case in which $P$ has row/column clustering with entries bounded away from 0.

As for the partially observed matrix $X$, we assume that its values are bounded, regardless of which entries are revealed (so our assumption will be on $X^*$, the version of $X$ that is fully-observed):

**A3.** There exists a constant $\phi \in (0, \infty)$ such that $\|X^*\|_{\max} \leq \phi$.

For example, in a recommendation systems context where $X$ is the ratings matrix, Assumption A3 holds if the ratings fall within a closed range of values (such as like/dislike where $X_{u,i} \in \{+1, -1\}$ and $\phi = 1$, or a rating out of five stars where $X_{u,i} \in [1, 5]$ and $\phi = 5$).

For simplicity, we do not place assumptions on signal matrix $S$ or noise matrix $W$ aside from their sum $X^*$ having bounded entries. Different assumptions on $S$ and $W$ lead to different matrix completion algorithms. Many of these algorithms can be debiased using estimated propensity scores. We focus our theoretical analysis on this debiasing step and experimentally apply the debiasing to a variety of matrix completion algorithms. We remark that there are existing papers that discuss how to handle MNAR data when $S$ is low rank and $W$ consists of i.i.d. zero-mean Gaussian (or sub-Gaussian) noise, a setup related to principal component analysis (e.g., Sportisse et al. [2018, 2019], Zhu et al. [2019]; a comparative study is provided by Dray and Josse [2015]).

Our main result is as follows. We defer the proof to Appendix B.

**Theorem 1.** *Under Assumptions A1–A3, suppose that we run algorithm* 1BITMC *with user-specified parameters satisfying $\tau \geq \theta$ and $\gamma \geq \alpha$ to obtain the estimate $\widehat{P}$ of propensity score matrix $P$. Let $\widehat{S} \in \mathbb{R}^{m \times n}$ be any matrix satisfying $\|\widehat{S}\|_{\max} \leq \psi$ for some $\psi \geq \phi$. Let $\delta \in (0, 1)$. Then there exists a universal constant $C > 0$ such that provided that $m + n \geq C$, with probability at least $1 - \frac{C}{m+n} - \delta$ over randomness in which entries are revealed in $X$, we simultaneously have*

$$\frac{1}{mn} \sum_{u=1}^{m} \sum_{i=1}^{n} (\widehat{P}_{u,i} - P_{u,i})^2 \leq 4e\tau \Big( \frac{1}{\sqrt{m}} + \frac{1}{\sqrt{n}} \Big), \tag{4}$$

$$|L_{\text{IPS-MSE}}(\widehat{S}|\widehat{P}) - L_{\text{full MSE}}(\widehat{S})| \leq \frac{8\psi^2\sqrt{e\tau}}{\sigma(-\gamma)\sigma(-\alpha)} \Big( \frac{1}{m^{1/4}} + \frac{1}{n^{1/4}} \Big) + \frac{4\psi^2}{\sigma(-\alpha)} \sqrt{\frac{1}{2mn} \log \frac{2}{\delta}}. \tag{5}$$

This theorem implies that under Assumptions A1–A3, with the number of rows and columns going to infinity, the IPS loss $L_{\text{IPS-MSE}}(\widehat{S}|\widehat{P})$ with $\widehat{P}$ computed using the 1BITMC algorithm is a consistent estimator for the fully-observed MSE loss $L_{\text{full MSE}}(\widehat{S})$.

We remark that our result easily extends to using MAE instead of MSE. If we define

$$L_{\text{full MAE}}(\widehat{S}) := \frac{1}{mn} \sum_{u=1}^{m} \sum_{i=1}^{n} |\widehat{S}_{u,i} - X^*_{u,i}|, \quad L_{\text{IPS-MAE}}(\widehat{S}|P) := \frac{1}{mn} \sum_{(u,i) \in \Omega} \frac{|\widehat{S}_{u,i} - X_{u,i}|}{P_{u,i}},$$

then the MAE version of Theorem 1 would replace (5) with

$$|L_{\text{IPS-MAE}}(\widehat{S}|\widehat{P}) - L_{\text{full MAE}}(\widehat{S})| \leq \frac{4\psi\sqrt{e\tau}}{\sigma(-\gamma)\sigma(-\alpha)} \Big( \frac{1}{m^{1/4}} + \frac{1}{n^{1/4}} \Big) + \frac{2\psi}{\sigma(-\alpha)} \sqrt{\frac{1}{2mn} \log \frac{2}{\delta}}.$$

Equations (2) and (3) both correspond to convex programs that can be efficiently solved via proximal gradient methods [Parikh and Boyd, 2014]. Hence we can find a $\widehat{S}$ that minimizes $L_{\text{IPS-MSE}}(\widehat{S}|\widehat{P})$, and it is straightforward to show that when $m, n \to \infty$, this $\widehat{S}$ also minimizes $L_{\text{full MSE}}(\widehat{S})$ since $|L_{\text{IPS-MSE}}(\widehat{S}|\widehat{P}) - L_{\text{full MSE}}(\widehat{S})| \to 0$.

## 4 Experiments

We now assess how well 1BITMC debiases matrix completion algorithms on synthetic and real data.

### 4.1 Synthetic Data

**Data.** We create two synthetic datasets that are intentionally catered toward propensity scores being well-explained by naive Bayes and logistic regression. 1) `MovieLoverData`: the dataset comes from the Movie-Lovers toy example (Figure 1 in Schnabel et al. [2016], which is based on Table 1 of Steck [2010]), where we set parameter $p = 0.5$; 2) `UserItemData`: for the second dataset, the "true" rating matrix and propensity score matrix are generated by the following steps.

We generate $U_1 \in [0,1]^{m \times 20}, V_1 \in [0,1]^{n \times 20}$ by sampling entries i.i.d. from Uniform$[0,1]$, and then form the form $\widetilde{S} = U_1 V_1^\top$. We scale the values of $\widetilde{S}$ to be from 1 to 5 and round to the nearest integer to produce the true ratings matrix $S$. Next, we generate row and column feature vectors $U_2 \in \mathbb{R}^{m \times 20}, V_2 \in \mathbb{R}^{n \times 20}$ by sampling entries i.i.d. from a normal distribution $\mathcal{N}(0, 1/64)$. We further generate $w_1 \in [0,1]^{20 \times 1}, w_2 \in [0,1]^{20 \times 1}$ by sampling entries i.i.d. from Uniform$[0,1]$. Then we form the propensity score matrix $P \in [0,1]^{m \times n}$ by setting $P_{u,i} = \sigma(U_2[u]w_1 + V_2[i]w_2)$, where $\sigma$ is the standard logistic function, and $U_2[u]$ denotes the $u$-th row of $U_2$. For both datasets, we set $m = 200, n = 300$. We also assume that i.i.d noise $\mathcal{N}(0,1)$ is added to each matrix entry of signal matrix $S$ in producing the partially revealed matrix $X$. All the ratings are clipped to $[1,5]$ and rounded. By sampling based on $P$, we generate $\Omega$, the training set indices. The true ratings matrix $S$ is used for testing. We briefly explain why Assumptions A1–A3 hold for these two datasets in Appendix D.

**Algorithms comparison.** We compare two types of algorithms for matrix completion. The first type does not account for entries being MNAR. This type of algorithm includes Probabilistic Matrix Factorization (PMF) [Mnih and Salakhutdinov, 2008], Funk's SVD [Funk, 2006], SVD++ [Koren, 2008], and SOFTIMPUTE [Mazumder et al., 2010]. The second type accounts for MNAR entries and includes max-norm-constrained matrix completion (MAXNORM) [Cai and Zhou, 2016], EXPOMF [Liang et al., 2016], and weighted-trace-norm-regularized matrix completion (WTN) [Srebro and Salakhutdinov, 2010]. For all the algorithms above (except for EXPOMF), the ratings in the squared error loss can be debiased by the propensity scores (as shown in equation (1)), and the propensity scores can be estimated from logistic regression (LR) (which requires extra user or item feature data), naive Bayes (NB) (specifically equation (18) of Schnabel et al. [2016], which requires a small set of missing at random (MAR) ratings), and 1BITMC [Davenport et al., 2014]. Hence we have a series of weighted-variants of the existing algorithms. For example, 1BITMC-PMF means the PMF method is used and the inverse propensity scores estimated from 1BITMC is used as weights for debiasing.

**Metrics.** We use MSE and MAE to measure the estimation quality of the propensity scores. Similarly, we also use MSE and MAE to compare the estimated full rating matrix with the true rating matrix $S$ (denoted as full-MSE or full-MAE). We also report SNIPS-MSE (SNIPS-MAE); these are evaluated on test set entries (i.e., all matrix entries in these synthetic datasets) using the true $P$.

**Experiment setup.** For all algorithms, we tune hyperparameters through 5-fold cross-validation using grid search. For the debiased methods (LR-⋆, NB-⋆, 1BITMC-⋆), we first estimate the propensity score matrix and then optimize the debiased loss. We note that `MovieLoverData` does not contain user/item features. Thus, naive Bayes can be used to estimate $P$ for `MovieLoverData` and `UserItemData`, while logistic regression is only applicable for `UserItemData`. In using logistic regression to estimate propensity scores, we can use all user/item features, only user features, or only item features (denoted as LR, LR-U, LR-I, respectively). Per dataset, we generate $P$ and $S$ once before generating 10 samples of noisy revealed ratings $X$ based on $P$ and $S$. We apply all the algorithms stated above to these 10 experimental repeats.

**Results.** Before looking at the performance of matrix completion methods, we first inspect the accuracy of the estimated propensity scores. Since we know the true propensity score for the synthetic datasets, we can compare the true $P$ with the estimated $\widehat{P}$ directly, as presented in Table 1. In how we constructed the synthetic datasets, unsurprisingly estimating propensity scores using naive Bayes on `MovieLoverData` and logistic regression on `UserItemData` achieve the best performance. In both cases, 1BITMC still achieves reasonably low errors in estimating the propensity score matrices.

| Algorithm | MovieLoverData | | UserItemData | |
|---|---|---|---|---|
| | MSE | MAE | MSE | MAE |
| Naive Bayes | **0.0346 ± 0.0002** | **0.1665 ± 0.0007** | 0.0150 ± 0.0001 | 0.0990 ± 0.0005 |
| LR | N/A | N/A | **0.0002 ± 0.0001** | **0.0105 ± 0.0017** |
| LR-U | N/A | N/A | 0.0070 ± 0.0000 | 0.0667 ± 0.0002 |
| LR-I | N/A | N/A | 0.0065 ± 0.0000 | 0.0639 ± 0.0001 |
| 1BITMC | 0.0520 ± 0.0003 | 0.1724 ± 0.0006 | 0.0119 ± 0.0000 | 0.0881 ± 0.0002 |

Table 1: Estimation accuracy of propensity score matrix (average ± standard deviation across 10 experimental repeats).

| Algorithm | MovieLoverData | | UserItemData | |
|---|---|---|---|---|
| | MSE | SNIPS-MSE | MSE | SNIPS-MSE |
| PMF | $0.326 \pm 0.042$ | $0.325 \pm 0.041$ | $0.161 \pm 0.002$ | $0.160 \pm 0.002$ |
| NB-PMF | $0.363 \pm 0.013$ | $0.363 \pm 0.012$ | $0.144 \pm 0.002$ | $0.145 \pm 0.002$ |
| LR-PMF | N/A | N/A | $0.159 \pm 0.002$ | $0.164 \pm 0.003$ |
| 1BITMC-PMF | $\mathbf{0.299 \pm 0.014}$ | $\mathbf{0.299 \pm 0.013}$ | $0.146 \pm 0.002$ | $0.146 \pm 0.002$ |
| SVD | $1.359 \pm 0.033$ | $1.360 \pm 0.034$ | $\mathbf{0.139 \pm 0.001}$ | $\mathbf{0.139 \pm 0.001}$ |
| NB-SVD | $0.866 \pm 0.028$ | $0.866 \pm 0.027$ | $0.147 \pm 0.001$ | $0.147 \pm 0.002$ |
| LR-SVD | N/A | N/A | $0.147 \pm 0.001$ | $0.152 \pm 0.002$ |
| 1BITMC-SVD | $0.861 \pm 0.028$ | $0.862 \pm 0.028$ | $\mathbf{0.139 \pm 0.001}$ | $\mathbf{0.139 \pm 0.001}$ |
| SVD++ | $0.343 \pm 0.023$ | $0.343 \pm 0.021$ | $0.140 \pm 0.001$ | $0.140 \pm 0.001$ |
| NB-SVD++ | $0.968 \pm 0.020$ | $0.987 \pm 0.020$ | $0.152 \pm 0.002$ | $0.153 \pm 0.002$ |
| LR-SVD++ | N/A | N/A | $0.154 \pm 0.001$ | $0.160 \pm 0.002$ |
| 1BITMC-SVD++ | $0.345 \pm 0.023$ | $0.345 \pm 0.021$ | $0.140 \pm 0.001$ | $0.140 \pm 0.001$ |
| SOFTIMPUTE | $0.374 \pm 0.009$ | $0.374 \pm 0.008$ | $0.579 \pm 0.002$ | $0.556 \pm 0.003$ |
| NB-SOFTIMPUTE | $0.495 \pm 0.010$ | $0.495 \pm 0.009$ | $0.599 \pm 0.004$ | $0.588 \pm 0.004$ |
| LR-SOFTIMPUTE | N/A | N/A | $0.602 \pm 0.003$ | $0.581 \pm 0.004$ |
| 1BITMC-SOFTIMPUTE | $0.412 \pm 0.011$ | $0.412 \pm 0.010$ | $0.588 \pm 0.002$ | $0.564 \pm 0.003$ |
| MAXNORM | $0.674 \pm 0.052$ | $0.674 \pm 0.053$ | $0.531 \pm 0.002$ | $0.507 \pm 0.002$ |
| NB-MAXNORM | $0.371 \pm 0.050$ | $0.371 \pm 0.049$ | $0.541 \pm 0.006$ | $0.520 \pm 0.007$ |
| LR-MAXNORM | N/A | N/A | $0.544 \pm 0.004$ | $0.521 \pm 0.005$ |
| 1BITMC-MAXNORM | $0.396 \pm 0.036$ | $0.395 \pm 0.035$ | $0.542 \pm 0.003$ | $0.519 \pm 0.004$ |
| WTN | $3.791 \pm 0.032$ | $3.790 \pm 0.035$ | $0.551 \pm 0.002$ | $0.528 \pm 0.002$ |
| NB-WTN | $3.262 \pm 0.093$ | $3.262 \pm 0.094$ | $0.557 \pm 0.002$ | $0.535 \pm 0.002$ |
| LR-WTN | N/A | N/A | $0.553 \pm 0.002$ | $0.532 \pm 0.002$ |
| 1BITMC-WTN | $3.788 \pm 0.039$ | $3.787 \pm 0.042$ | $0.551 \pm 0.002$ | $0.528 \pm 0.002$ |
| EXPOMF | $0.820 \pm 0.005$ | $0.822 \pm 0.005$ | $1.170 \pm 0.008$ | $1.218 \pm 0.009$ |

Table 2: MSE-based metrics of matrix completion methods on synthetic datasets (average $\pm$ standard deviation across 10 experimental repeats).

Now we compare the matrix completion methods directly and report the performance of different methods in Table 2. Note that we only show the MSE-based results; the MAE-based results are presented in Appendix D. The debiased variants generally perform as well as or better than their original unmodified counterparts. 1BITMC-PMF achieves the best accuracy on MovieLoverData, and both SVD and 1BITMC-SVD perform the best on UserItemData. The debiasing using 1BITMC can improve the performance of PMF, SVD, MAXNORM and WTN on MovieLoverData, and PMF is improved on UserItemData. In general, debiasing using 1BITMC leads to higher matrix completion accuracy than debiasing using LR and NB.

## 4.2 Real-World Data

**Data.** We consider two real-world datasets. 1) Coat: the dataset contains ratings from 290 users on 300 items [Schnabel et al., 2016]. The dataset contains both MNAR ratings as well as MAR ratings. Both user and item features are available for the dataset. 2) MovieLens-100k: the dataset contains 100k ratings from 943 users on 1,682 movies, and it does not contain any MAR ratings [Harper and Konstan, 2016].

**Experiments setup.** Since the Coat dataset contains both MAR and MNAR data, we are able to train the algorithms on the MNAR data and test on the MAR data. In this way, the MSE (MAE) on the testing set directly reflect the matrix completion accuracy. For MovieLens-100k, we split the data into 90/10 train/test sets 10 times. For both datasets, we use 5-fold cross-validation to tune the hyperparameters through grid search. The SNIPS related measures are computed on test data based on the propensities estimated from 1BITMC-PMF using training data.

| Algorithm | Coat | | MovieLens-100k | |
|---|---|---|---|---|
| | MSE | SNIPS-MSE | MSE | SNIPS-MSE |
| PMF | 1.000 | **1.051** | $0.896 \pm 0.013$ | $0.902 \pm 0.013$ |
| NB-PMF | 1.034 | 1.117 | N/A | N/A |
| LR-PMF | 1.025 | 1.107 | N/A | N/A |
| 1BITMC-PMF | 0.999 | 1.052 | $0.845 \pm 0.012$ | $0.853 \pm 0.011$ |
| SVD | 1.203 | 1.270 | $0.862 \pm 0.013$ | $0.872 \pm 0.012$ |
| NB-SVD | 1.246 | 1.346 | N/A | N/A |
| LR-SVD | 1.234 | 1.334 | N/A | N/A |
| 1BITMC-SVD | 1.202 | 1.272 | $\mathbf{0.821 \pm 0.011}$ | $\mathbf{0.832 \pm 0.011}$ |
| SVD++ | 1.208 | 1.248 | $0.838 \pm 0.013$ | $0.849 \pm 0.012$ |
| NB-SVD++ | 1.488 | 1.608 | N/A | N/A |
| LR-SVD++ | 1.418 | 1.532 | N/A | N/A |
| 1BITMC-SVD++ | 1.248 | 1.274 | $0.833 \pm 0.012$ | $0.843 \pm 0.011$ |
| SOFTIMPUTE | 1.064 | 1.150 | $0.929 \pm 0.015$ | $0.950 \pm 0.015$ |
| NB-SOFTIMPUTE | 1.052 | 1.138 | N/A | N/A |
| LR-SOFTIMPUTE | 1.069 | 1.156 | N/A | N/A |
| 1BITMC-SOFTIMPUTE | **0.998** | 1.078 | $0.933 \pm 0.014$ | $0.953 \pm 0.014$ |
| MAXNORM | 1.168 | 1.263 | $0.911 \pm 0.011$ | $0.925 \pm 0.011$ |
| NB-MAXNORM | 1.460 | 1.578 | N/A | N/A |
| LR-MAXNORM | 1.537 | 1.662 | N/A | N/A |
| 1BITMC-MAXNORM | 1.471 | 1.590 | $0.977 \pm 0.017$ | $0.992 \pm 0.019$ |
| WTN | 1.396 | 1.509 | $0.939 \pm 0.013$ | $0.952 \pm 0.013$ |
| NB-WTN | 1.329 | 1.437 | N/A | N/A |
| LR-WTN | 1.340 | 1.448 | N/A | N/A |
| 1BITMC-WTN | 1.396 | 1.509 | $0.934 \pm 0.013$ | $0.946 \pm 0.013$ |
| EXPOMF | 2.602 | 2.813 | $2.461 \pm 0.077$ | $2.558 \pm 0.083$ |

Table 3: MSE-based metrics of matrix completion methods on `Coat` and `MovieLens-100k` (results for `MovieLens-100k` are the averages $\pm$ standard deviations across 10 experimental repeats).

**Results.** The performance of each algorithm is presented in Table 3. We report the MSE-based results; MAE results are in Appendix D. Algorithms 1BITMC-SOFTIMPUTE and 1BITMC-PMF perform the best on `Coat` based on MSE, and 1BITMC-SVD outperforms the rest on `MovieLens-100k`. The debiasing approach does not improve the accuracy for MAXNORM and WTN.

# 5 Conclusions

In this paper, we examined the effectiveness of debiasing matrix completion algorithms using missingness probabilities (propensity scores) estimated via another matrix completion algorithm: 1BITMC by Davenport et al. [2014], which relies on low nuclear norm structure, and which we apply to a fully-revealed missingness mask matrix (so we are doing matrix *denoising* rather than completion). Our numerical experiments indicate that debiasing using 1BITMC can achieve downstream matrix completion accuracy at least as good as debiasing using logistic regression and naive Bayes baselines, despite 1BITMC not using auxiliary information such as row/column feature vectors. Moreover, debiasing matrix completion algorithms with 1BITMC can boost accuracy, in some cases achieving the best or nearly the best performance across all algorithms we tested. These experimental findings suggest that a low nuclear norm assumption on missingness patterns is reasonable.

In terms of theoretical analysis, we have not addressed the full generality of MNAR data in matrix completion. For example, we still assume that each entry is revealed independent of other entries. In reality, one matrix entry being revealed could increase (or decrease) the chance of another entry being revealed. As another loose end, our theory breaks down when a missingness probability is exactly 0. For example, consider when the matrix to be completed corresponds to feature vectors collected from patients. A clinical measurement that only makes sense for women will have 0 probability of being revealed for men. In such scenarios, imputing such a missing value does not actually make sense. These are two open problems among many for robustly handling MNAR data with guarantees.

## Footnotes

[2]Note that $L_{\text{IPS-MSE}}(\widehat{S}|P) = \frac{1}{mn} \sum_{u=1}^{m} \sum_{i=1}^{n} \mathbb{1}\{(u,i) \in \Omega\} \frac{(\widehat{S}_{u,i} - X^*_{u,i})^2}{P_{u,i}}$. Taking the expectation with respect to which entries are revealed, $\mathbb{E}_\Omega[L_{\text{IPS-MSE}}(\widehat{S}|P)] = \frac{1}{mn} \sum_{u=1}^{m} \sum_{i=1}^{n} P_{u,i} \frac{(\widehat{S}_{u,i} - X^*_{u,i})^2}{P_{u,i}} = L_{\text{full MSE}}(\widehat{S})$.

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
