[Supplementary Material · mnar_mc_supplemental.pdf]

# Supplemental Material

## A  Topic Modeling Structure

Not only do the `Coat` and `MovieLens-100k` datasets have block structure, they can also be explained using topic modeling structure, which is again a low rank model for the propensity score matrix $P$ (and is thus a special case of the general low nuclear norm structure we assume provided that $P$ be bounded away from 0). We build on our biclustering data exploration example from Figure 1. Specifically, in the `MovieLens-100k` dataset, each user can be thought of as a distribution over movie "topics", and each movie topic corresponds to a distribution over movie genres, as shown in Figure 2. Meanwhile, in the `Coat` dataset, each item can be represented as a distribution over user topics, and each user topic can be thought of as a distribution over user features, as shown in Figure 3.

Figure 2: Average item features for each movie topic in `MovieLens-100k`.

Figure 3: Average user features for each user topic in `Coat`.

Topic models reveal co-occurrence information. For example, movie topic 1 in Figure 2 corresponds to users who tend to reveal ratings for action, adventure, musical, sci-fi, and war movies, but not documentaries. Movie topic 4 corresponds to users who tend to reveal ratings for mysteries and thrillers. Users can be associated with multiple topics to varying degrees. Ratings for documentaries tend to mostly be revealed by users associated with movie topic 0. We can also find such patterns in the `Coat` dataset, where we look at topics over rows instead of columns as an illustration. We can see what sorts of user features tend to be co-occur. For example, user topic 2 consists of men uninterested in fashion. Coats can be associated with different user topics to varying degrees.

Note that the biclustering uses only the missingness information. We are interpreting the biclustering results here with the help of user/item features to identify co-occurrence relationships.

# B  Proof of Theorem 1

Denote the objective function (which is a log likelihood) of optimization problem (3) as

$$L_M(\Gamma) := \sum_{u=1}^{m} \sum_{i=1}^{n} [M_{u,i} \log \sigma(\Gamma_{u,i}) + (1 - M_{u,i}) \log(1 - \sigma(\Gamma_{u,i}))].$$

We specialize Theorem 2 of Davenport et al. [2014] to the setting in which all entries of the matrix are observed. Their proof makes use of an offset version of the log likelihood function:

$$\overline{L}_M(\Gamma) := L_M(\Gamma) - L_M(\mathbf{0}) = \sum_{u=1}^{m} \sum_{i=1}^{n} \left[ M_{u,i} \log \frac{\sigma(\Gamma_{u,i})}{\sigma(0)} + (1 - M_{u,i}) \log \frac{1 - \sigma(\Gamma_{u,i})}{1 - \sigma(0)} \right].$$

We let $\mathcal{F}_{\tau,\gamma}$ denote the feasible set of optimization problem (3), i.e.,

$$\mathcal{F}_{\tau,\gamma} := \left\{ \Gamma \in \mathbb{R}^{m \times n} \; : \; \|\Gamma\|_* \leq \tau \sqrt{mn}, \; \|\Gamma\|_{\max} \leq \gamma \right\}.$$

Moreover, we actually let the function $\sigma$ be a bit more general: $\sigma$ must be differentiable, and the following quantity must exist and be finite:

$$L_\gamma := \sup_{x \in [-\gamma, \gamma]} \frac{|\sigma'(x)|}{\sigma(x)(1 - \sigma(x))}. \tag{6}$$

We prove the following more general theorem.

**Theorem 2.** *Under Assumptions A1–A3, suppose that we run algorithm* 1BITMC *with user-specified parameters satisfying $\tau \geq \theta$ and $\gamma \geq \alpha$ to obtain the estimate $\widehat{P}$ of propensity score matrix $P$. Let $\widehat{S} \in \mathbb{R}^{m \times n}$ be any matrix satisfying $\|\widehat{S}\|_{\max} \leq \psi$ for some $\psi \geq \phi$. Let $\delta \in (0, 1)$. Then there exists a universal constant $C > 0$ such that provided that $m + n \geq C$, with probability at least $1 - \frac{C}{m+n} - \delta$ over randomness in which entries are revealed in X, we simultaneously have*

$$\frac{1}{mn} \sum_{u=1}^{m} \sum_{i=1}^{n} (\widehat{P}_{u,i} - P_{u,i})^2 \leq 4eL_\gamma \tau \left( \frac{1}{\sqrt{m}} + \frac{1}{\sqrt{n}} \right), \tag{7}$$

$$|L_{\text{IPS-MSE}}(\widehat{S}|\widehat{P}) - L_{\text{full MSE}}(\widehat{S})| \leq \frac{8\psi^2 \sqrt{eL_\gamma \tau}}{\sigma(-\gamma)\sigma(-\alpha)} \left( \frac{1}{m^{1/4}} + \frac{1}{n^{1/4}} \right) + \frac{4\psi^2}{\sigma(-\alpha)} \sqrt{\frac{1}{2mn} \log \frac{2}{\delta}}. \tag{8}$$

We recover Theorem 1 by noting that for $\sigma$ chosen to be the standard logistic function, we have $L_\gamma = 1$ for all $\gamma > 0$.

## Proof of Theorem 2

Under Assumption A1 and since $\tau \geq \theta$, note that $A \in \mathcal{F}_{\tau,\gamma}$. By optimality of $\widehat{A}$ for optimization problem (3), we have $L_M(\widehat{A}) \geq L_M(A)$, which can written as

$$0 \leq L_M(\widehat{A}) - L_M(A)$$
$$= \overline{L}_M(\widehat{A}) - \overline{L}_M(A)$$
$$= (\overline{L}_M(\widehat{A}) - \mathbb{E}_M[\overline{L}_M(\widehat{A})]) - (\overline{L}_M(A) - \mathbb{E}_M[\overline{L}_M(A)]) + \mathbb{E}_M[\overline{L}_M(\widehat{A}) - \overline{L}_M(A)].$$

Since matrices $\widehat{A}$ and $A$ are both in the set $\mathcal{F}_{\tau,\gamma}$, the first two terms on the right-hand side can each be upper-bounded by $\sup_{\Gamma \in \mathcal{F}_{\tau,\gamma}} |\overline{L}_M(\Gamma) - \mathbb{E}_M[\overline{L}_M(\Gamma)]|$. Meanwhile, the third term on the right-hand

side can be rewritten as

$$\mathbb{E}_M[\overline{L}_M(\widehat{A}) - \overline{L}_M(A)] = \sum_{u=1}^{m} \sum_{i=1}^{n} \mathbb{E}_{M_{u,i}} \Big[ M_{u,i} \log \frac{\sigma(\widehat{A}_{u,i})}{\sigma(A_{u,i})} + (1 - M_{u,i}) \log \frac{1 - \sigma(\widehat{A}_{u,i})}{1 - \sigma(A_{u,i})} \Big]$$

$$= \sum_{u=1}^{m} \sum_{i=1}^{n} \Big[ P_{u,i} \log \frac{\sigma(\widehat{A}_{u,i})}{\sigma(A_{u,i})} + (1 - P_{u,i}) \log \frac{1 - \sigma(\widehat{A}_{u,i})}{1 - \sigma(A_{u,i})} \Big]$$

$$= \sum_{u=1}^{m} \sum_{i=1}^{n} \Big[ P_{u,i} \log \frac{\widehat{P}_{u,i}}{P_{u,i}} + (1 - P_{u,i}) \log \frac{1 - \widehat{P}_{u,i}}{1 - P_{u,i}} \Big]$$

$$= - \sum_{u=1}^{m} \sum_{i=1}^{n} \Big[ P_{u,i} \log \frac{P_{u,i}}{\widehat{P}_{u,i}} + (1 - P_{u,i}) \log \frac{1 - P_{u,i}}{1 - \widehat{P}_{u,i}} \Big]$$

$$= - \sum_{u=1}^{m} \sum_{i=1}^{n} D\big(\mathrm{Ber}(P_{u,i}) \| \mathrm{Ber}(\widehat{P}_{u,i})\big).$$

Putting together the pieces, we have

$$0 \le (\overline{L}_M(\widehat{A}) - \mathbb{E}_M[\overline{L}_M(\widehat{A})]) - (\overline{L}_M(A) - \mathbb{E}_M[\overline{L}_M(A)]) + \mathbb{E}_M[\overline{L}_M(\widehat{A}) - \overline{L}_M(A)]$$

$$\le 2 \sup_{\Gamma \in \mathcal{F}_{\tau,\gamma}} |\overline{L}_M(\Gamma) - \mathbb{E}_M[\overline{L}_M(\Gamma)]| + \mathbb{E}_M[\overline{L}_M(\widehat{A}) - \overline{L}_M(A)]$$

$$= 2 \sup_{\Gamma \in \mathcal{F}_{\tau,\gamma}} |\overline{L}_M(\Gamma) - \mathbb{E}_M[\overline{L}_M(\Gamma)]| - \sum_{u=1}^{m} \sum_{i=1}^{n} D\big(\mathrm{Ber}(P_{u,i}) \| \mathrm{Ber}(\widehat{P}_{u,i})\big). \qquad (9)$$

By Pinsker's inequality,

$$D\big(\mathrm{Ber}(P_{u,i}) \| \mathrm{Ber}(\widehat{P}_{u,i})\big) \ge 2\|\mathrm{Ber}(P_{u,i}) - \mathrm{Ber}(\widehat{P}_{u,i})\|_{\mathrm{TV}}^2$$

$$= 2 \Big[ \frac{1}{2} (|P_{u,i} - \widehat{P}_{u,i}| + |(1 - P_{u,i}) - (1 - \widehat{P}_{u,i})|) \Big]^2$$

$$= 2(\widehat{P}_{u,i} - P_{u,i})^2.$$

Therefore,

$$\sum_{u=1}^{m} \sum_{i=1}^{n} D\big(\mathrm{Ber}(P_{u,i}) \| \mathrm{Ber}(\widehat{P}_{u,i})\big) \ge 2 \sum_{u=1}^{m} \sum_{i=1}^{n} (\widehat{P}_{u,i} - P_{u,i})^2. \qquad (10)$$

Combining inequalities (9) and (10), we get

$$\sum_{u=1}^{m} \sum_{i=1}^{n} (\widehat{P}_{u,i} - P_{u,i})^2 \le \sup_{\Gamma \in \mathcal{F}_{\tau,\gamma}} |\overline{L}_M(\Gamma) - \mathbb{E}_M[\overline{L}_M(\Gamma)]|.$$

The next lemma upper-bounds $\sup_{\Gamma \in \mathcal{F}_{\tau,\gamma}} |\overline{L}_M(\Gamma) - \mathbb{E}_M[\overline{L}_M(\Gamma)]|$.

**Lemma 3.** *For the above setup, if $m + n \ge 3$, then there exists a universal constant $C > 0$ such that*

$$\mathbb{P}\Big( \sup_{\Gamma \in \mathcal{F}_{\tau,\gamma}} |\overline{L}_M(\Gamma) - \mathbb{E}_M[\overline{L}_M(\Gamma)]| \ge 4eL_\gamma \tau \sqrt{mn}(\sqrt{m} + \sqrt{n}) \Big) \le \frac{C}{m + n}. \qquad (11)$$

Once this lemma is established, the theorem's first main inequality (7) readily follows since with probability at least $1 - \frac{C}{m+n}$ (for which we clearly want $m + n \ge C$),

$$\frac{1}{mn} \sum_{u=1}^{m} \sum_{i=1}^{n} (\widehat{P}_{u,i} - P_{u,i})^2 \le \frac{1}{mn} [4eL_\gamma \tau \sqrt{mn}(\sqrt{m} + \sqrt{n})] = 4eL_\gamma \tau \Big( \frac{1}{\sqrt{m}} + \frac{1}{\sqrt{n}} \Big),$$

which establishes inequality (4). Note that Lemma 3 asks that $m + n \ge 3$. Since $C = 8 \cdot 2^{1/4} \cdot e^2 = 70.2969\ldots$, asking that $m + n \ge C$ implies that $m + n \ge 3$.

We now derive the theorem's second main inequality (8), which is a consequence of the first main inequality (7). By the triangle inequality,

$$|L_{\text{IPS-MSE}}(\widehat{S}|\widehat{P}) - L_{\text{full MSE}}(\widehat{S})|$$
$$\leq |L_{\text{IPS-MSE}}(\widehat{S}|\widehat{P}) - L_{\text{IPS-MSE}}(\widehat{S}|P)| + |L_{\text{IPS-MSE}}(\widehat{S}|P) - L_{\text{full MSE}}(\widehat{S})|. \qquad (12)$$

We can readily bound the first RHS term as follows:

$$|L_{\text{IPS-MSE}}(\widehat{S}|\widehat{P}) - L_{\text{IPS-MSE}}(\widehat{S}|P)| = \left| \frac{1}{mn} \sum_{(u,i) \in \Omega} \left[ \frac{(\widehat{S}_{u,i} - X_{u,i})^2}{\widehat{P}_{u,i}} - \frac{(\widehat{S}_{u,i} - X_{u,i})^2}{P_{u,i}} \right] \right|$$

$$= \left| \frac{1}{mn} \sum_{(u,i) \in \Omega} \frac{P_{u,i} - \widehat{P}_{u,i}}{\widehat{P}_{u,i} P_{u,i}} (\widehat{S}_{u,i} - X_{u,i})^2 \right|$$

$$\text{worst case error } |\widehat{S}_{u,i} - X_{u,i}| \leq 2\psi \quad \leq \frac{4\psi^2}{mn} \sum_{(u,i) \in \Omega} \frac{|P_{u,i} - \widehat{P}_{u,i}|}{\widehat{P}_{u,i} P_{u,i}}$$

$$\text{note that } \widehat{P}_{u,i} \geq \sigma(-\gamma) \text{ and } P_{u,i} \geq \sigma(-\alpha) \quad \leq \frac{4\psi^2}{\sigma(-\gamma)\sigma(-\alpha)mn} \sum_{(u,i) \in \Omega} |P_{u,i} - \widehat{P}_{u,i}|$$

$$\text{basic inequality relating } \ell_1 \text{ and } \ell_2 \text{ norms} \quad \leq \frac{4\psi^2}{\sigma(-\gamma)\sigma(-\alpha)mn} \sqrt{|\Omega|} \sqrt{\sum_{(u,i) \in \Omega} (\widehat{P}_{u,i} - P_{u,i})^2}$$

$$= \frac{4\psi^2}{\sigma(-\gamma)\sigma(-\alpha)} \sqrt{\frac{|\Omega|}{mn}} \sqrt{\frac{1}{mn} \sum_{(u,i) \in \Omega} (\widehat{P}_{u,i} - P_{u,i})^2}$$

$$\text{the theorem's main inequality (7)} \quad \leq \frac{4\psi^2}{\sigma(-\gamma)\sigma(-\alpha)} \sqrt{\frac{|\Omega|}{mn}} \sqrt{4eL_\gamma \tau \left( \frac{1}{\sqrt{m}} + \frac{1}{\sqrt{n}} \right)}$$

$$\text{fraction of observed entries } \tfrac{|\Omega|}{mn} \text{ is at most 1} \quad \leq \frac{4\psi^2}{\sigma(-\gamma)\sigma(-\alpha)} \sqrt{4eL_\gamma \tau \left( \frac{1}{\sqrt{m}} + \frac{1}{\sqrt{n}} \right)}$$

$$= \frac{8\psi^2 \sqrt{eL_\gamma \tau}}{\sigma(-\gamma)\sigma(-\alpha)} \sqrt{\frac{1}{\sqrt{m}} + \frac{1}{\sqrt{n}}}$$

$$\leq \frac{8\psi^2 \sqrt{eL_\gamma \tau}}{\sigma(-\gamma)\sigma(-\alpha)} \left( \frac{1}{m^{1/4}} + \frac{1}{n^{1/4}} \right),$$

where the last step uses the fact that $(a+b)^p \leq a^p + b^p$ for all $p \in [0,1]$ and $a, b \in \mathbb{R}_+$.

The second RHS term in inequality (12) can be bounded with Hoeffding's inequality. First, recall that

$$L_{\text{IPS-MSE}}(\widehat{S}|P) = \frac{1}{mn} \sum_{u=1}^m \sum_{i=1}^n \mathbb{1}\{(u,i) \in \Omega\} \frac{(\widehat{S}_{u,i} - X^*_{u,i})^2}{P_{u,i}},$$

which is an average of $mn$ random variables (here, the only randomness we are considering is in which entries are revealed $\Omega$). One can check that $\mathbb{E}_\Omega[L_{\text{IPS-MSE}}(\widehat{S}|P)] = L_{\text{full MSE}}(\widehat{S})$. Note that $\frac{(\widehat{S}_{u,i} - X^*_{u,i})^2}{P_{u,i}} \leq \frac{(2\psi)^2}{\sigma(-\alpha)} = \frac{4\psi^2}{\sigma(-\alpha)}$. Thus, each of the terms in the double summation above is bounded in the interval $[0, \frac{4\psi^2}{\sigma(-\alpha)}]$, so by Hoeffding's inequality,

$$\mathbb{P}\left( |L_{\text{IPS-MSE}}(\widehat{S}|P) - L_{\text{full MSE}}(\widehat{S})| \geq \frac{4\psi^2}{\sigma(-\alpha)} \sqrt{\frac{1}{2mn} \log \frac{2}{\delta}} \right) \leq \delta. \qquad (13)$$

When this bad event does not happen, then the second RHS term of triangle inequality (12) is at most $\frac{4\psi^2}{\sigma(-\alpha)} \sqrt{\frac{1}{2mn} \log \frac{2}{\delta}}$, so putting together the pieces, we get the theorem's second main inequality (8). By a union bound, the bad events corresponding to bounds (11) and (13) both don't happen with probability at least $1 - \frac{C}{m+n} - \delta$.

*Proof of Lemma 3.* By Markov's inequality, for any $h > 0$ and $z > 0$, we have

$$\mathbb{P}\Big(\sup_{\Gamma \in \mathcal{F}_{\tau,\gamma}} |\overline{L}_M(\Gamma) - \mathbb{E}_M[\overline{L}_M(\Gamma)]| \geq z\Big) = \mathbb{P}\Big(\sup_{\Gamma \in \mathcal{F}_{\tau,\gamma}} |\overline{L}_M(\Gamma) - \mathbb{E}_M[\overline{L}_M(\Gamma)]|^h \geq z^h\Big)$$

$$\leq \frac{\mathbb{E}_M\big[\sup_{\Gamma \in \mathcal{F}_{\tau,\gamma}} |\overline{L}_M(\Gamma) - \mathbb{E}_M[\overline{L}_M(\Gamma)]|^h\big]}{z^h}. \quad (14)$$

We will be setting $h = \log(m+n)$ (which is greater than 1 under the assumption that $m + n \geq 3$) and $z = 4eL_\gamma \tau \sqrt{mn}(\sqrt{m} + \sqrt{n})$.

We next upper-bound the numerator term $\mathbb{E}_M\big[\sup_{\Gamma \in \mathcal{F}_{\tau,\gamma}} |\overline{L}_M(\Gamma) - \mathbb{E}_M[\overline{L}_M(\Gamma)]|^h\big]$. To do this, we apply a standard symmetrization argument. Let matrix $M' \in \mathbb{R}^{m \times n}$ be independently sampled such that $M'$ and $M$ have the same distribution. Then using Jensen's inequality,

$$\mathbb{E}_M\Big[\sup_{\Gamma \in \mathcal{F}_{\tau,\gamma}} |\overline{L}_M(\Gamma) - \mathbb{E}_M[\overline{L}_M(\Gamma)]|^h\Big] = \mathbb{E}_M\Big[\sup_{\Gamma \in \mathcal{F}_{\tau,\gamma}} |\overline{L}_M(\Gamma) - \mathbb{E}_M[\overline{L}_M(\Gamma)]|^h\Big]$$

$$= \mathbb{E}_M\Big[\sup_{\Gamma \in \mathcal{F}_{\tau,\gamma}} |\overline{L}_M(\Gamma) - \mathbb{E}_{M'}[\overline{L}_{M'}(\Gamma)]|^h\Big]$$

$$= \mathbb{E}_M\Big[\sup_{\Gamma \in \mathcal{F}_{\tau,\gamma}} |\mathbb{E}_{M'}[\overline{L}_M(\Gamma) - \overline{L}_{M'}(\Gamma)]|^h\Big]$$

$$\leq \mathbb{E}_M\Big[\mathbb{E}_{M'}\Big[\sup_{\Gamma \in \mathcal{F}_{\tau,\gamma}} |\overline{L}_M(\Gamma) - \overline{L}_{M'}(\Gamma)|^h\Big]\Big]$$

$$= \mathbb{E}_{M,M'}\Big[\sup_{\Gamma \in \mathcal{F}_{\tau,\gamma}} |\overline{L}_M(\Gamma) - \overline{L}_{M'}(\Gamma)|^h\Big]. \quad (15)$$

In applying Jensen's inequality, we are using the fact that as a function of $M'$, the function $\sup_{\Gamma \in \mathcal{F}_{\tau,\gamma}} |\overline{L}_M(\Gamma) - \overline{L}_{M'}(\Gamma)|^h$ (for $h \geq 1$) is the pointwise supremum of convex functions, so it is still convex.

Next, we examine the random variable $\overline{L}_M(\Gamma) - \overline{L}_{M'}(\Gamma)$. We shall be introducing independently sampled Rademacher random variables $\xi_{u,i} \in \{\pm 1\}$ for $u \in [m]$ and $i \in [n]$. Note that

$$\overline{L}_M(\Gamma) - \overline{L}_{M'}(\Gamma) = \sum_{u=1}^{m} \sum_{i=1}^{n} \Big[M_{u,i} \log \frac{\sigma(\Gamma_{u,i})}{\sigma(0)} + (1 - M_{u,i}) \log \frac{1 - \sigma(\Gamma_{u,i})}{1 - \sigma(0)}\Big]$$

$$- \sum_{u=1}^{m} \sum_{i=1}^{n} \Big[M'_{u,i} \log \frac{\sigma(\Gamma_{u,i})}{\sigma(0)} + (1 - M'_{u,i}) \log \frac{1 - \sigma(\Gamma_{u,i})}{1 - \sigma(0)}\Big]$$

$$= \sum_{u=1}^{m} \sum_{i=1}^{n} \Big[M_{u,i} \log \frac{\sigma(\Gamma_{u,i})}{\sigma(0)} + (1 - M_{u,i}) \log \frac{1 - \sigma(\Gamma_{u,i})}{1 - \sigma(0)}$$

$$- M'_{u,i} \log \frac{\sigma(\Gamma_{u,i})}{\sigma(0)} - (1 - M'_{u,i}) \log \frac{1 - \sigma(\Gamma_{u,i})}{1 - \sigma(0)}\Big]$$

has the same distribution as the random variable

$$\overline{L}_M(\Gamma) - \overline{L}_{M'}(\Gamma) = \sum_{u=1}^{m} \sum_{i=1}^{n} \xi_{u,i} \Big[M_{u,i} \log \frac{\sigma(\Gamma_{u,i})}{\sigma(0)} + (1 - M_{u,i}) \log \frac{1 - \sigma(\Gamma_{u,i})}{1 - \sigma(0)}$$

$$- M'_{u,i} \log \frac{\sigma(\Gamma_{u,i})}{\sigma(0)} - (1 - M'_{u,i}) \log \frac{1 - \sigma(\Gamma_{u,i})}{1 - \sigma(0)}\Big].$$

Then, using the fact that $|a + b|^p \leq 2^{p-1}(|a|^p + |b|^p)$ for $p \geq 1$ and $a, b \in \mathbb{R}$,

$$\mathbb{E}_{M,M'}\left[ \sup_{\Gamma \in \mathcal{F}_{\tau,\gamma}} |\overline{L}_M(\Gamma) - \overline{L}_{M'}(\Gamma)|^h \right]$$

$$= \mathbb{E}_{M,M'}\left[ \sup_{\Gamma \in \mathcal{F}_{\tau,\gamma}} \left| \sum_{u=1}^m \sum_{i=1}^n \xi_{u,i}\left[ M_{u,i} \log \frac{\sigma(\Gamma_{u,i})}{\sigma(0)} + (1 - M_{u,i}) \log \frac{1 - \sigma(\Gamma_{u,i})}{1 - \sigma(0)} \right. \right. \right.$$
$$\left. \left. \left. - M'_{u,i} \log \frac{\sigma(\Gamma_{u,i})}{\sigma(0)} - (1 - M'_{u,i}) \log \frac{1 - \sigma(\Gamma_{u,i})}{1 - \sigma(0)} \right] \right|^h \right]$$

$$\leq 2^{h-1} \mathbb{E}_{M,M'}\left[ \sup_{\Gamma \in \mathcal{F}_{\tau,\gamma}} \left( \left| \sum_{u=1}^m \sum_{i=1}^n \xi_{u,i}\left[ M_{u,i} \log \frac{\sigma(\Gamma_{u,i})}{\sigma(0)} + (1 - M_{u,i}) \log \frac{1 - \sigma(\Gamma_{u,i})}{1 - \sigma(0)} \right] \right|^h \right. \right.$$
$$\left. \left. + \left| \sum_{u=1}^m \sum_{i=1}^n \xi_{u,i}\left[ M'_{u,i} \log \frac{\sigma(\Gamma_{u,i})}{\sigma(0)} + (1 - M'_{u,i}) \log \frac{1 - \sigma(\Gamma_{u,i})}{1 - \sigma(0)} \right] \right|^h \right) \right]$$

$$\leq 2^{h-1} \left( \mathbb{E}_{M,M'}\left[ \sup_{\Gamma \in \mathcal{F}_{\tau,\gamma}} \left| \sum_{u=1}^m \sum_{i=1}^n \xi_{u,i}\left[ M_{u,i} \log \frac{\sigma(\Gamma_{u,i})}{\sigma(0)} + (1 - M_{u,i}) \log \frac{1 - \sigma(\Gamma_{u,i})}{1 - \sigma(0)} \right] \right|^h \right] \right.$$
$$\left. + \mathbb{E}_{M,M'}\left[ \sup_{\Gamma \in \mathcal{F}_{\tau,\gamma}} \left| \sum_{u=1}^m \sum_{i=1}^n \xi_{u,i}\left[ M'_{u,i} \log \frac{\sigma(\Gamma_{u,i})}{\sigma(0)} + (1 - M'_{u,i}) \log \frac{1 - \sigma(\Gamma_{u,i})}{1 - \sigma(0)} \right] \right|^h \right] \right)$$

$$= 2^h \mathbb{E}_M\left[ \sup_{\Gamma \in \mathcal{F}_{\tau,\gamma}} \left| \sum_{u=1}^m \sum_{i=1}^n \xi_{u,i}\left[ M_{u,i} \log \frac{\sigma(\Gamma_{u,i})}{\sigma(0)} + (1 - M_{u,i}) \log \frac{1 - \sigma(\Gamma_{u,i})}{1 - \sigma(0)} \right] \right|^h \right]. \qquad (16)$$

At this point, we use a contraction argument. As a reminder,

$$L_\gamma := \sup_{x \in [-\gamma, \gamma]} \frac{|\sigma'(x)|}{\sigma(x)(1 - \sigma(x))}.$$

Thus, for any $x \in [-\gamma, \gamma]$,

$$\left| \frac{d}{dx} \log \frac{\sigma(x)}{\sigma(0)} \right| = \left| \frac{d}{dx} \log \sigma(x) \right| = \left| \frac{1}{\sigma(x)} \sigma'(x) \right| \leq \left| \frac{\sigma'(x)}{\sigma(x)(1 - \sigma(x))} \right| \leq L_\gamma,$$

i.e., $x \mapsto \log \frac{\sigma(x)}{\sigma(0)}$ is $L_\gamma$-Lipschitz. A similar argument can be used to justify that $x \mapsto \log \frac{1 - \sigma(x)}{1 - \sigma(0)}$ is $L_\gamma$-Lipschitz over $[-\gamma, \gamma]$. Hence, $x \mapsto \frac{1}{L_\gamma} \log \frac{\sigma(x)}{\sigma(0)}$ and $x \mapsto \frac{1}{L_\gamma} \log \frac{1 - \sigma(x)}{1 - \sigma(0)}$ are both contractions (i.e., 1-Lipschitz). Applying the second inequality of Theorem 11.6 by Boucheron et al. [2013] (with $\Psi(x) := x^h$) and defining $\overline{M}_{u,i} = 2M_{u,i} - 1 \in \{\pm 1\}$,

$$\mathbb{E}_M\left[ \sup_{\Gamma \in \mathcal{F}_{\tau,\gamma}} \left| \sum_{u=1}^m \sum_{i=1}^n \xi_{u,i}\left[ M_{u,i} \log \frac{\sigma(\Gamma_{u,i})}{\sigma(0)} + (1 - M_{u,i}) \log \frac{1 - \sigma(\Gamma_{u,i})}{1 - \sigma(0)} \right] \right|^h \right]$$

$$= \mathbb{E}_M\left[ \Psi\left( \sup_{\Gamma \in \mathcal{F}_{\tau,\gamma}} \left| \sum_{u=1}^m \sum_{i=1}^n \xi_{u,i}\left[ M_{u,i} \log \frac{\sigma(\Gamma_{u,i})}{\sigma(0)} + (1 - M_{u,i}) \log \frac{1 - \sigma(\Gamma_{u,i})}{1 - \sigma(0)} \right] \right| \right) \right]$$

$$= (2L_\gamma)^h \mathbb{E}_M\left[ \Psi\left( \frac{1}{2L_\gamma} \sup_{\Gamma \in \mathcal{F}_{\tau,\gamma}} \left| \sum_{u=1}^m \sum_{i=1}^n \xi_{u,i}\left[ M_{u,i} \log \frac{\sigma(\Gamma_{u,i})}{\sigma(0)} + (1 - M_{u,i}) \log \frac{1 - \sigma(\Gamma_{u,i})}{1 - \sigma(0)} \right] \right| \right) \right]$$

$$\leq (2L_\gamma)^h \mathbb{E}_M\left[ \Psi\left( \sup_{\Gamma \in \mathcal{F}_{\tau,\gamma}} \left| \sum_{u=1}^m \sum_{i=1}^n \xi_{u,i}\left[ M_{u,i}\Gamma_{u,i} - (1 - M_{u,i})\Gamma_{u,i} \right] \right| \right) \right]$$

$$= (2L_\gamma)^h \mathbb{E}_M\left[ \Psi\left( \sup_{\Gamma \in \mathcal{F}_{\tau,\gamma}} \left| \sum_{u=1}^m \sum_{i=1}^n \xi_{u,i}\overline{M}_{u,i}\Gamma_{u,i} \right| \right) \right]$$

$$= (2L_\gamma)^h \mathbb{E}_M\left[ \sup_{\Gamma \in \mathcal{F}_{\tau,\gamma}} \left| \sum_{u=1}^m \sum_{i=1}^n \xi_{u,i}\overline{M}_{u,i}\Gamma_{u,i} \right|^h \right]$$

$$= (2L_\gamma)^h \mathbb{E}_M\Big[ \sup_{\Gamma \in \mathcal{F}_{\tau,\gamma}} |\langle \Xi \circ \overline{M}, \Gamma\rangle|^h\Big], \tag{17}$$

where $\Xi \in \{\pm 1\}^{m \times n}$ has its $(u,i)$-th entry given by $\xi_{u,i}$, "$\circ$" denotes the Hadamard product, and $\langle \cdot, \cdot \rangle$ denotes the trace inner product.

Next, we use the result that $|\langle A, B\rangle| \leq \|A\|_2 \|B\|_*$, so

$$\mathbb{E}_M\Big[ \sup_{\Gamma \in \mathcal{F}_{\tau,\gamma}} |\langle \Xi \circ \overline{M}, \Gamma\rangle|^h\Big] \leq \mathbb{E}_M\Big[ \sup_{\Gamma \in \mathcal{F}_{\tau,\gamma}} \|\Xi \circ \overline{M}\|_2^h \|\Gamma\|_*^h\Big]$$

$$\leq \mathbb{E}_M\Big[ \sup_{\Gamma \in \mathcal{F}_{\tau,\gamma}} \|\Xi \circ \overline{M}\|_2^h (\alpha\sqrt{rmn})^h\Big]$$

$$= (\tau\sqrt{mn})^h \mathbb{E}_M[\|\Xi \circ \overline{M}\|_2^h]. \tag{18}$$

Finally, applying Theorem 1.1 of Seginer [2000], there exists a universal constant $C > 0$ such that

$$\mathbb{E}_M[\|\Xi \circ \overline{M}\|_2^h] \leq C(m^{h/2} + n^{h/2}). \tag{19}$$

In fact, $C = 8 \cdot 2^{1/4} \cdot e^2 = 70.2969\ldots$

At this point, stringing together inequalities (15), (16), (17), (18), and (19), we get

$$\mathbb{E}_M\Big[ \sup_{\Gamma \in \mathcal{F}_{\tau,\gamma}} |\overline{L}_M(\Gamma) - \mathbb{E}_M[\overline{L}_M(\Gamma)]|^h\Big]$$

$$\leq \mathbb{E}_{M,M'}\Big[ \sup_{\Gamma \in \mathcal{F}_{\tau,\gamma}} |\overline{L}_M(\Gamma) - \overline{L}_{M'}(\Gamma)|^h\Big]$$

$$\leq 2^h \mathbb{E}_M\Big[ \sup_{\Gamma \in \mathcal{F}_{\tau,\gamma}} \Big| \sum_{u=1}^m \sum_{i=1}^n \xi_{u,i}\Big[M_{u,i}\log\frac{\sigma(\Gamma_{u,i})}{\sigma(0)} + (1-M_{u,i})\log\frac{1-\sigma(\Gamma_{u,i})}{1-\sigma(0)}\Big]\Big|^h\Big]$$

$$\leq 2^h(2L_\gamma)^h \mathbb{E}_M\Big[ \sup_{\Gamma \in \mathcal{F}_{\tau,\gamma}} |\langle \Xi \circ \overline{M}, \Gamma\rangle|^h\Big]$$

$$\leq 2^h(2L_\gamma)^h(\tau\sqrt{mn})^h \mathbb{E}_M[\|\Xi \circ \overline{M}\|_2^h]$$

$$\leq 2^h(2L_\gamma)^h(\tau\sqrt{mn})^h C(m^{h/2} + n^{h/2})$$

$$= C(4L_\gamma\tau\sqrt{mn})^h(m^{h/2} + n^{h/2}).$$

Lastly, using the fact that $(a + b)^p \leq a^p + b^p$ for $p \in [0,1]$ and $a, b \in \mathbb{R}_+$

$$\Big(\mathbb{E}_M\Big[ \sup_{\Gamma \in \mathcal{F}_{\tau,\gamma}} |\overline{L}_M(\Gamma) - \mathbb{E}_M[\overline{L}_M(\Gamma)]|^h\Big]\Big)^{1/h} \leq [C(4L_\gamma\tau\sqrt{mn})^h(m^{h/2} + n^{h/2})]^{1/h}$$

$$\leq C^{1/h}4L_\gamma\tau\sqrt{mn}(\sqrt{m} + \sqrt{n}).$$

Finally, by choosing $h = \log(m+n)$ and $z = 4eL_\gamma\tau\sqrt{mn}(\sqrt{m} + \sqrt{n})$ in inequality (14),

$$\mathbb{P}\Big( \sup_{\Gamma \in \mathcal{F}_{\tau,\gamma}} |\overline{L}_M(\Gamma) - \mathbb{E}_M[\overline{L}_M(\Gamma)]| \geq z\Big) \leq \frac{\mathbb{E}_M\big[ \sup_{\Gamma \in \mathcal{F}_{\tau,\gamma}} |\overline{L}_M(\Gamma) - \mathbb{E}_M[\overline{L}_M(\Gamma)]|^h\big]}{z^h}.$$

$$\leq \frac{[C^{1/\log(m+n)}4L_\gamma\tau\sqrt{mn}(\sqrt{m} + \sqrt{n})]^{\log(m+n)}}{[4eL_\gamma\tau\sqrt{mn}(\sqrt{m} + \sqrt{n})]^{\log(m+n)}}$$

$$= \frac{C}{m+n}. \qquad \square$$

## C  Modifying 1BITMC to Allow for Propensity Scores of 1

We now discuss how to modify 1BITMC along with its theoretical analysis to allow for entries in the propensity score matrix $P$ to be 1. It suffices to make a single change to the algorithm: we replace the feasible set $\mathcal{F}_{\tau,\gamma}$ in optimization problem (3) by

$$\mathcal{F}_{\tau,\gamma,\varphi} := \big\{ \Gamma \in \mathbb{R}^{m \times n} : \|\Gamma\|_* \leq \tau\sqrt{mn},$$

$$\Gamma_{u,i} \geq -\gamma \text{ for all } u, i,$$

$$\Gamma_{u,i} \leq \varphi \text{ for all } u, i \text{ s.t. } M_{u,i} = 0\big\}, \tag{20}$$

where we have introduced a new user-specified parameter $\varphi \in (-\gamma, \gamma)$. The resulting modified optimization problem is still convex if the original optimization program was convex (which depends on the choice of $\sigma$). The key idea for the modification is that we allow $\Gamma_{u,i}$ to be as large as possible for entries where $M_{u,i} = 1$ (to allow for $\sigma(\Gamma_{u,i}) = 1$, assuming that $\sigma$ monotonically increases to 1). However, when $M_{u,i} = 0$, we enforce that $\Gamma_{u,i}$ cannot be too large. For example, if the $j$-th column is always observed ($M_{u,j} = 1$ for all $u$), then there would be no upper bound constraint on any element in the $j$-th column of $\Gamma$.

For completeness, we present this modified version of 1BITMC in Algorithm 1, which we call 1BITMC-MODIFIED; note that we now intentionally use $\Sigma$ rather than $\sigma$ to denote the link function to avoid confusion as we will take $\sigma$ to be the standard logistic function and $\Sigma$ to a be different function in our theoretical analysis.

---

**Algorithm 1:** 1BITMC-MODIFIED

---

**Data:** Binary matrix $M \in \{0,1\}^{m \times n}$, nuclear norm constraint parameter $\tau > 0$, lower bound parameter $\gamma > 0$, upper bound parameter $\varphi > -\gamma$, function $\Sigma : \mathbb{R} \to [0,1]$ (maps real number to probability)

**Result:** Estimate $\widehat{P}$ of $P$

1 Solve optimization problem (3) with feasible set $\mathcal{F}_{\tau,\gamma}$ replaced by $\mathcal{F}_{\tau,\gamma,\varphi}$ as given in equation (20).
2 Set $\widehat{P}_{u,i} := \sigma(\widehat{A}_{u,i})$ for all $u \in [m], i \in [n]$.

---

**Theoretical Analysis**

How the theory changes is more involved. A key theoretical consequence of using feasible set $\mathcal{F}_{\tau,\gamma,\varphi}$ is that we will only be able to accurate estimate entries of $P$ that are in the set $[\Sigma(-\gamma), \Sigma(\varphi)] \cup \{1\}$. We tolerate error in estimating entries of $P$ that are in the "critical" interval $(\Sigma(\varphi), 1)$ (with $\varphi$ chosen to be sufficiently large, this interval length could be made arbitrarily small). We denote the fraction of entries in $P$ that are in the critical interval as

$$f_{\text{critical}}(m, n) := \frac{1}{mn} \sum_{u=1}^{m} \sum_{i=1}^{n} \mathbb{1}\{P_{u,i} \in (\Sigma(\varphi), 1)\}.$$

We no longer assume that the true propensity score matrix $P$ is linked to parameter matrix $A$ via the standard logistic function $\sigma$. Instead, we reparameterize $P$ via $P_{u,i} = \Sigma(A_{u,i})$, where

$$\Sigma(x) := \begin{cases} \sigma(x) & \text{for } x < -\gamma, \\ \sigma(x) + \underbrace{\frac{1}{2}\left(1 + \frac{x}{\gamma}\right)(1 - \sigma(\gamma))}_{\substack{\text{linear correction term that is} \\ 0 \text{ at } x = -\gamma \text{ and } 1 - \sigma(\gamma) \text{ at } x = \gamma}} & \text{for } x \in [-\gamma, \gamma], \\ 1 & \text{for } x > \gamma. \end{cases} \tag{21}$$

The above choice of $\Sigma$ depends on algorithm parameter $\gamma$. Observe that $A_{u,i} \geq \gamma$ implies that $P_{u,i} = 1$ (previously when linking with the standard logistic function, we could not achieve $P_{u,i} = 1$ for a finite $A_{u,i}$ value). Our theoretical guarantee for 1BITMC-MODIFIED depends on the following quantity that summarizes Lipschitz smoothness information involving $\log \Sigma$ and $\log(1 - \Sigma)$:

$$\Upsilon_{\gamma,\varphi} := \max\left\{1 + \frac{1}{2\gamma}, \frac{1}{1 - \Sigma(\varphi)}\left(\frac{1}{4} + \frac{1}{2\gamma}\right)\right\}.$$

Next, we replace Assumption A2 with the following much more general assumption:

**A2′.** There exists some $p_{\min} > 0$ such that $P_{u,i} \geq p_{\min}$ for all $u \in [m]$ and $i \in [n]$.

We are now ready to state our theoretical guarantee for 1BITMC-MODIFIED.

**Theorem 4.** *Under Assumptions A1, A2′, and A3, suppose that we run algorithm 1BITMC-MODIFIED with $\Sigma$ as given in equation (21), $\tau \geq \theta$, $\Sigma(-\gamma) \leq p_{\min}$, and $\varphi \in (-\gamma, \gamma)$ to obtain the estimate $\widehat{P}$ of propensity score matrix $P$. Let $\widehat{S} \in \mathbb{R}^{m \times n}$ be any matrix satisfying $\|\widehat{S}\|_{\max} \leq \psi$ for some $\psi \geq \phi$.*

*Let $\delta \in (0, 1)$. Then there exists a universal constant $C > 0$ such that provided that $m + n \geq C$, with probability at least $1 - \frac{C}{m+n} - \delta$ over randomness in which entries are revealed in $X$, we simultaneously have*

$$\frac{1}{mn} \sum_{u=1}^{m} \sum_{i=1}^{n} (\widehat{P}_{u,i} - P_{u,i})^2 \leq 8e \Upsilon_{\gamma,\varphi} \tau \left( \frac{1}{\sqrt{m}} + \frac{1}{\sqrt{n}} \right) + \frac{(1 - \Sigma(\varphi))^2}{2} f_{critical}(m, n), \quad (22)$$

$$|L_{IPS\text{-}MSE}(\widehat{S}|\widehat{P}) - L_{full\,MSE}(\widehat{S})| \leq \frac{4\psi^2}{\Sigma(-\gamma)p_{\min}} \left[ \sqrt{8e \Upsilon_{\gamma,\varphi} \tau} \left( \frac{1}{m^{1/4}} + \frac{1}{n^{1/4}} \right) \right.$$
$$\left. + (1 - \Sigma(\varphi)) \sqrt{\frac{f_{critical}(m, n)}{2}} \right]$$
$$+ \frac{4\psi^2}{p_{\min}} \sqrt{\frac{1}{2mn} \log \frac{2}{\delta}}. \quad (23)$$

Ignoring terms involving $f_{critical}(m, n)$, the two bounds (22) and (23) are qualitatively similar to their counterparts in Theorem 1 (our main result for 1BITMC). The $f_{critical}(m, n)$ terms are approximation errors in choosing algorithm parameter $\varphi$ poorly. If there exists some constant $p_{critical} \in (p_{\min}, 1)$ such that $P_{u,i} \in [p_{\min}, p_{critical}] \cup \{1\}$ for all $u \in [m], i \in [n]$, and $\varphi \in (-\gamma, \gamma)$ is chosen so that $\Sigma(\varphi) \geq p_{critical}$, then observe that $f_{critical}(m, n) = 0$. In general, if $f_{critical}(m, n)$ is nonzero, then we can still have the two error bounds go to 0 provided as $m, n \to \infty$, we have $f_{critical}(m, n) \to 0$.

**Proof of Theorem 4**

The theorem is a consequence of the following lemma, which we sketch a proof for at the end of this section.

**Lemma 5.** *Under the same assumptions as in Theorem 4, further assume that there exists $p_{critical} \in (p_{min}, 1)$ such that $P_{u,i} \in [p_{\min}, p_{critical}] \cup \{1\}$ for all $u \in [m], i \in [n]$, and $\varphi \in (-\gamma, \gamma)$ is chosen so that $\Sigma(\varphi) \geq p_{critical}$. Then there exists a universal constant $C > 0$ such that provided that $m + n \geq C$, with probability at least $1 - \frac{C}{m+n}$ over randomness in which entries are revealed in $X$, we have*

$$\frac{1}{mn} \sum_{u=1}^{m} \sum_{i=1}^{n} (\widehat{P}_{u,i} - P_{u,i})^2 \leq 4e \Upsilon_{\gamma,\varphi} \tau \left( \frac{1}{\sqrt{m}} + \frac{1}{\sqrt{n}} \right). \quad (24)$$

In general, $P$ might not satisfy the additional assumption in Lemma 5. What we do is consider the projection of $P$ onto matrices that do satisfy the additional assumption. Namely, let the projection $P^\dagger \in [0, 1]^{m \times n}$ be defined as

$$P_{u,i}^\dagger = \begin{cases} P_{u,i} & \text{if } P_{u,i} \in [p_{\min}, \Sigma(\varphi)] \cup 1, \\ \Sigma(\varphi) & \text{if } P_{u,i} \in \left( \Sigma(\varphi), \frac{\Sigma(\varphi)+1}{2} \right], \\ 1 & \text{if } P_{u,i} \in \left( \frac{\Sigma(\varphi)+1}{2}, 1 \right]. \end{cases}$$

Matrix $P^\dagger$ is guaranteed to satisfy the conditions on the propensity score matrix in Lemma 5 with $p_{critical} = \Sigma(\varphi)$.

Next, we have

$$\frac{1}{mn} \|\widehat{P} - P\|_F^2 \leq \frac{1}{mn} (\|\widehat{P} - P^\dagger\|_F + \|P^\dagger - P\|_F)^2$$
$$\leq \frac{2}{mn} \|\widehat{P} - P^\dagger\|_F^2 + \frac{2}{mn} \|P^\dagger - P\|_F^2. \quad (25)$$

We can bound the first RHS term using Lemma 5:

$$\frac{2}{mn} \|\widehat{P} - P^\dagger\|_F^2 \leq 8e \Upsilon_{\gamma,\varphi} \tau \left( \frac{1}{\sqrt{m}} + \frac{1}{\sqrt{n}} \right). \quad (26)$$

The second RHS term in inequality (25) can be upper-bounded by noticing that the worst-case absolute entry-wise error of $\frac{1-\Sigma(\varphi)}{2}$ occurs only for $u \in [m], i \in [n]$ such that $P_{u,i} \in (\Sigma(\varphi), 1)$.

Thus,

$$\frac{2}{mn}\|P^\dagger - P\|_F^2 \leq \frac{2}{mn}\left(\frac{1-\Sigma(\varphi)}{2}\right)^2 \sum_{u=1}^{m}\sum_{i=1}^{n} \mathbf{1}\{P_{u,i} \in (\Sigma(\varphi), 1)\}$$

$$= \frac{(1-\Sigma(\varphi))^2}{2} f_{\text{critical}}(m,n). \tag{27}$$

Combining inequalities (25), (26), and (27) yields the theorem's first main bound (22). The theorem's second main bound (23) can then be established using the same proof ideas as in establishing bound (8) for Theorem 2.

**Proof Sketch for Lemma 5**

We highlight the main change to the proof of bound (7) in Theorem 2. Specifically, we do not assume the condition given by equation (6) that defines the variable $L_\gamma$ (in fact, as we explain shortly, we replace $L_\gamma$ with $\Upsilon_{\gamma,\varphi}$). This affects the contraction argument made in inequality (17). At the start of inequality (17) (for which we replace $\sigma$ with $\Sigma$), each term of the summation has a factor

$$M_{u,i}\log\frac{\Sigma(\Gamma_{u,i})}{\Sigma(0)} + (1 - M_{u,i})\log\frac{1-\Sigma(\Gamma_{u,i})}{1-\Sigma(0)}. \tag{28}$$

Exactly one of the two terms can be nonzero since $M_{u,i} \in \{0,1\}$. Previously, we showed that $x \mapsto \log\frac{\sigma(x)}{\sigma(0)}$ and $x \mapsto \log\frac{1-\sigma(x)}{1-\sigma(0)}$ were each $L_\gamma$-Lipschitz for $x \in [-\gamma, \gamma]$ (i.e., $x \mapsto \frac{1}{L_\gamma}\log\frac{\sigma(x)}{\sigma(0)}$ and $x \mapsto \frac{1}{L_\gamma}\log\frac{1-\sigma(x)}{1-\sigma(0)}$ are contractions). Now we show the analogous result using $\Sigma$ instead of $\sigma$ and with the new feasible set $\mathcal{F}_{\tau,\gamma,\varphi}$. There are two cases to consider:

**Case 1 ($M_{u,i} = 1$).** The only possibly nonzero term in expression (28) is $\log\frac{\Sigma(\Gamma_{u,i})}{\Sigma(0)}$, where $\Gamma_{u,i} \in [-\gamma, \gamma]$. We show that the function $x \mapsto \frac{1}{\Upsilon_{\gamma,\varphi}}\log\frac{\Sigma(x)}{\Sigma(0)}$ (for $x \in [-\gamma, \gamma]$) is a contraction by showing that $|\frac{d}{dx}\log\frac{\Sigma(x)}{\Sigma(0)}| \leq \Upsilon_{\gamma,\varphi}$. Recall that the standard logistic function $\sigma$ has $\frac{|\sigma'(x)|}{\sigma(x)(1-x)} = 1$. Also, by construction, $\Sigma(x) \geq \sigma(x)$. We have, for $x \in [-\gamma, \gamma]$,

$$\left|\frac{d}{dx}\log\frac{\Sigma(x)}{\Sigma(0)}\right| = \left|\frac{d}{dx}\log\Sigma(x)\right|$$

$$= \frac{1}{\Sigma(x)}\cdot\left[\sigma'(x) + \frac{1-\sigma(\gamma)}{2\gamma}\right]$$

$$\leq \frac{1}{\sigma(x)}\cdot\left[\sigma'(x) + \frac{1-\sigma(\gamma)}{2\gamma}\right]$$

$$\leq \frac{\sigma'(x)}{\sigma(x)(1-\sigma(x))} + \frac{1}{\sigma(x)}\cdot\frac{1-\sigma(\gamma)}{2\gamma}$$

$$= 1 + \frac{1}{\sigma(x)}\cdot\frac{1-\sigma(\gamma)}{2\gamma}$$

$$\leq 1 + \frac{1}{\sigma(-\gamma)}\cdot\frac{1-\sigma(\gamma)}{2\gamma}$$

$$= 1 + \frac{1}{2\gamma}$$

$$\leq \Upsilon_{\gamma,\varphi}.$$

**Case 2 ($M_{u,i} = 0$).** The only possibly nonzero term in expression (28) is $\log\frac{1-\Sigma(\Gamma_{u,i})}{1-\Sigma(0)}$, where $\Gamma_{u,i} \in [-\gamma, \varphi]$. We show that the function $x \mapsto \frac{1}{\Upsilon_{\gamma,\varphi}}\log\frac{1-\Sigma(x)}{1-\Sigma(0)}$ (for $x \in [-\gamma, \varphi]$) is a contraction by showing that $|\frac{d}{dx}\log\frac{1-\Sigma(x)}{1-\Sigma(0)}| \leq \Upsilon_{\gamma,\varphi}$. Note that $\sigma'(x) \leq 1/4$ for all $x \in \mathbb{R}$. We have, for $x \in [-\gamma, \varphi]$,

$$\left|\frac{d}{dx}\log\frac{1-\Sigma(x)}{1-\Sigma(0)}\right| = \left|\frac{d}{dx}\log(1-\Sigma(x))\right|$$

$$= \left| \frac{1}{1 - \Sigma(x)} \cdot \frac{d}{dx} (1 - \Sigma(x)) \right|$$

$$= \frac{1}{1 - \Sigma(x)} \cdot \left[ \sigma'(x) + \frac{1 - \sigma(\gamma)}{2\gamma} \right]$$

$$\leq \frac{1}{1 - \Sigma(\varphi)} \cdot \left[ \sigma'(x) + \frac{1 - \sigma(\gamma)}{2\gamma} \right]$$

$$\leq \frac{1}{1 - \Sigma(\varphi)} \cdot \left[ \frac{1}{4} + \frac{1}{2\gamma} \right]$$

$$\leq \Upsilon_{\gamma,\varphi}.$$

## D  More Details on Experiments

In this section, we explain why Assumptions A1–A3 hold for the two synthetic datasets (with high probability in the case of `UserItemData`), and we also present MAE-based results for the numerical experiments on both synthetic and real-world datasets.

### D.1  Sythetic Data

We verify that Assumptions A1-A3 hold for the synthetic datasets. Assumption A3 holds as both synthetic datasets have partially observed matrix $X$ consist of ratings in a bounded interval. For `MovieLoverData`, the propensity score matrix $P$ is a block matrix, so it is low-rank, and moreover it has three unique values that are all nonzero and less than 1; thus Assumptions A1 and A2 are both met. For `UserItemData`, the propensity score is $P_{u,i} = \sigma(A_{u,i})$ where $A_{u,i} = U_2[u]w_1 + V_2[i]w_2$ where $\sigma$ is the standard logistic function. Hence, parameter matrix $A$ (in Assumptions A1 and A2) has low rank, and in practice after we generate $A$ we can find what its maximum absolute value entry is to satisfy Assumption A2. Alternatively, to obtain a bound that holds with high probability, standard concentration inequality results for the maxima of a finite collection of sub-Gaussian random variables can be used to bound $\|A\|_{\max}$. In summary, the synthetic datasets we consider satisfy the assumptions of our theoretical analysis.

The MAE-based measures for different algorithms on `MovieLoverData` and `UserItemData` are presented in Table 4.

### D.2  Real-World Data

The MAE-based measures for different algorithms on `Coat` and `MovieLens-100k` are presented in Table 5.