[Reviews · NeurIPS 2019]

Reviewer 1



I have to say I really enjoyed reading this paper. The motivation, problem and insights are clear and well presented. The paper addresses the problem of de-biasing a non-uniform sampling pattern for matrix completion by using a clever existing technique (1bit matrix completion) to estimate the matrix P indicating the probability of observing each entry. My only concern is that this idea relies on such matrix P having low nuclear norm. Empirical evidence suggest that this is true for real datasets (experiments in the paper show that this technique produces a reasonable completion). However, it would be interesting to theoretically characterize (or at least provide some insights as to) which sampling patterns would correspond to a P with low nuclear norm. In lack of such theoretical characterization, and since in general (as far as I know) P is unknown for most (if not all) real datasets, one way to obtain insights that justify the assumptions would be to analyze the type of patterns produced by matrices P with low nuclear norm, and test whether they resemble real data sampling patterns.

Reviewer 2



Originality: According to lines 59-61, it seems to me that the proposed 1BITMC approach is just a special case of the approach proposed by Davenport et al. 2014. This paper also states that the 1BITMC approach is originally proposed by Davenport et al. 2014 (lines 114-115). In this sense, the paper does not propose any novel approaches. The theoretical results of the 1BITMC approach, which seem to be the main contributions of this paper, are mostly adapted from those of Davenport et al. 2014. Quality: I am a bit concerned about the experiment setup in Section 4.2. This paper randomly split the data into a training, a validation, and a testing set. This paper uses the testing set to evaluate all rating prediction approaches in terms of MAE and three variants of MAE-IPS. This evaluation procedure might be biased because the testing set is not missing at random and hence is biased. Instead, existing studies (Schnabel et al. 2016 and Wang et al. 2019) use missing at random ratings collected by forcing users to rate randomly selected items as the testing set. Clarity: This paper is mostly well written with a few exceptions. For example, the paper defines a noise matrix at line 77, but never models the rating noise or uses the noise matrix in the propensity estimation. So, it seems to me that the definition of the noise matrix is not necessary for the overall flow of the paper. Significance: This paper demonstrates that the proposed 1BITMC approach is significantly better than the naive Bayes and the logistic regression approach on synthetic datasets. However, the experimental results of Section 4.2 in the rating prediction and the classification task on the real datasets seem not to be significant to me.

Reviewer 3



Since the algorithm of estimating the propensity is proposed by Davenport et al. 2014, the originality of the paper mainly lies in the bounds derivation and experiments. For the bounds of the bias and overall completion error, there is no direct experiments bridging the proposed theory and practice. I would like more empirical evidences on the assumptions from real-world matrices, beyond the recommendation domain where COAT and MovieLens are from. The novelty of the paper is also less impressive when the motivation of investigating the adoption of nuclear norm is unclear. From the experiments, it is only demonstrated that the proposed propensity estimator can achieve similar results as previous classic methods (and can be even slightly worse if data fits better for Naive Bayes or Logistic Regression). The performance gain of the newly proposed estimator on the MovieLens dataset (the largest experimented datasets) is not very significant compared with Naive Bayes, meaning that when m and n are large the bias and completion error is similar to Naive Bayes. Admitted that the new estimator does not require more features or MAR data, I would still say the established knowledge from this paper is not very significant in its current form. It can do better by considering whether we can use the user/item features and the MAR data when we have them in the 1BitMC algorithm. Can we then largely improve the SOTA? The paper is in general well written and easy to follow. To make it self-contained, it would be better to introduce some background about nuclear norm. The authors are also encouraged to spent slightly fewer spaces on the background of IPS related approaches, and introduce a bit more on the 1-bit matrix completion algorithms since it is a closely related work and the algorithm is the working horse for the proposed estimator.

[Author Response · NeurIPS 2019]

We thank the reviewers for providing very thoughtful and helpful comments. In general, the reviews mainly focus on
three concerns: motivation for low nuclear norm structure (from all three reviewers), originality (from reviewers #2 and
#3), and experiment settings (from reviewers #2 and #3). We address these main concerns below.

**Motivation for low nuclear norm structure.** All three reviewers asked for us better relating the low nuclear norm
structure in missingness patterns to real data (reviewer #2 also asks about relating it to synthetic data). Importantly, we
want to reiterate that our assumptions of low nuclear norm (Assumption A1) and propensities scores being bounded
from 0 and from 1 (Assumption A2) have special cases that include low-rank propensity score matrices, which are a
broad class of matrices that the propensity scores can have (special cases include block-structured propensity score
matrices that would, for instance, come from row/column clustering, or as a more elaborate example, topic modeling
structure). We discuss why this low rank condition is a special case after stating Assumptions A1 and A2 in the paper.
To be concrete, for the synthetic datasets we consider, we can easily verify how Assumptions A1–A3 are satisfied. For
the real data we consider, we do not have the true propensity score matrices, but we can do the following: we can take
the missingness mask matrices and check for whether they have block structure, which would suggest that how entries
are revealed is approximately low rank. We discuss these in detail next; we will add these details to the paper.

*Synthetic datasets.* We verify that Assumptions A1-A3 hold for the synthetic datasets. Assumption A3 holds as both
synthetic datasets have partially observed matrix $X$ consist of ratings in a bounded interval. For `MovieLoverData`,
the propensity score matrix $P$ is a block matrix, so it is low-rank, and moreover it has three unique values that are
all nonzero and less than 1; thus Assumptions A1 and A2 are both met. For `UserItemData`, the propensity score is
$P_{u,i} = g(U_2[u]w_1 + V_2[i]w_2)$, where $g$ is the standard logistic function. Hence parameter matrix $A$ (in Assumptions A1
and A2) is low-rank and, moreover, in how we generate $A$, with high probability $\|A\|_{\max}$ is bounded (using standard
concentration results for the maxima of a finite collection of Gaussians); thus Assumptions A1 and A2 are both satisfied.
In summary, the synthetic datasets we consider do actually satisfy the assumptions of our theoretical analysis.

*Real-world data.* To demonstrate that real-world datasets we consider satisfy the low nuclear norm assumption, we run
the following experiments: we use bi-clustering algorithms to rearrange rows/columns of the missingness mask matrix
to reveal block structure; the results for `COAT` and `MovieLens` are presented in Figure 1.

Figure 1: Biclustering the missingness mask matrices. (a,b,c): Raw, biclustered missingness mask matrices and average missingness for `COAT`; (d,e,f): Raw, biclustered missingness mask matrices and average missingness for `MovieLens`.

One can see that block structure appears in both datasets, which suggests the propensity score matrix can be well
modeled as low-rank, which is a special case of the low nuclear norm (provided that propensity score is bounded from 0
and 1). We will add this figure to the paper.

**Originality.** Reviewer #2 mentioned that the propensity score estimation method 1BITMC is based on Davenport et
al. [2014], which is not developed by the authors. We agree with reviewer #2 that 1BITMC is not new. We specialize
their analysis to fully-observed matrices. This fully-observed setting allows us to arrive at a slightly tighter error upper
bound. Even so, we consider our theoretical analysis to be an incremental contribution. Instead, we argue that the main
contribution of our paper is to demonstrate that low nuclear norm structure in propensity score matrices appears to be
reasonable for real-world data and to provide a theoretically-justified propensity estimation approach when MAR data
and user/item features are not available. Though the idea is straightforward, we believe the proposed approach is novel
and useful for estimating the propensity scores without using any extra information. Reviewer #3 commented that the
novelty of this paper is undermined by insufficient motivation for low nuclear norm structure in missingness patterns.
We hope that we addressed this concern with the clarifications in previous paragraphs.

**Experiment settings.** Reviewer #2 believes that using MNAR ratings as the test set might be biased, while we argue
that the measure $L_{\text{IPS-MAE}}$ is unbiased even with MNAR test set. In both synthetic and real-world datasets, we compare
different algorithms through $L_{\text{IPS-MAE}}$ to avoid bias. One major contribution of this paper is to estimate the propensity
matrix without using MAR data or user/item features, hence we prefer to work on datasets that do not contain MAR
ratings. Instead, we believe using $L_{\text{IPS-MAE}}$ is enough to conduct the fair and unbiased evaluation. We would like
to thank reviewer #3 for the suggestion of using user/item features and MAR data together with 1BITMC to further
improve performance. We consider this a promising future research direction. For simplicity, in this paper we focus on
propensity score estimation without any extra information (but using nuclear norm structure) and benchmark against
logistic regression and naive Bayes baeslines that do use auxiliary information.

[Meta-Review · NeurIPS 2019]

This paper addresses the problem of handling missing not-at-random measurements in matrix completion. This is not a new line of thought in statistics literature, but this paper nicely bridges the ideas to present them to a NeurIPS audience. That said, it seems like the authors are unaware of some key recent work, including those I include below. In their revision, the reviewers must look into and include citations from this literature. Dray, Stéphane, and Julie Josse. "Principal component analysis with missing values: a comparative survey of methods." Plant Ecology 216, no. 5 (2015): 657-667. Steck, Harald. "Training and testing of recommender systems on data missing not at random." In Proceedings of the 16th ACM SIGKDD international conference on Knowledge discovery and data mining, pp. 713-722. ACM, 2010. and if the authors are willing to cite very new work that came out this summer: Sportisse, Aude, Claire Boyer, and Julie Josse. "Estimation and imputation in Probabilistic Principal Component Analysis with Missing Not At Random data." (2019). https://arxiv.org/abs/1906.02493